# A Technoeconomic Platform for Early-Stage Process Design and Cost Estimation of Joint Fermentative-Catalytic Bioprocessing

**Mothi Bharath Viswanathan [1,*], D. Raj Raman [1,*], Kurt A. Rosentrater [1] and Brent H. Shanks [2]**

1    Department of Agricultural and Biosystems Engineering, Iowa State University, Ames, IA 50011, USA;
    karosent@iastate.edu
2    Department of Chemical and Biological Engineering, Iowa State University, Ames, IA 50011, USA;
    bshanks@iastate.edu
*    Correspondence: mothiv@illinois.edu (M.B.V.); rajraman@iastate.edu (D.R.R.);
    Tel.: +515-203-7040 (M.B.V.); +515-294-0465 (D.R.R.)

**Abstract:** Technoeconomic analyses using established tools such as SuperPro Designer® require a level of detail that is typically unavailable at the early stage of process evaluation. To facilitate this, members of our group previously created a spreadsheet-based process modeling and technoeconomic platform explicitly aimed at joint fermentative-catalytic biorefinery processes. In this work, we detail the reorganization and expansion of this model—ESTEA2 (Early State Technoeconomic Analysis, version 2), including detailed design and cost calculations for new unit operations. Furthermore, we describe ESTEA2 validation using ethanol and sorbic acid process. The results were compared with estimates from the literature, SuperPro Designer® (Version 8.5, Intelligen Inc., Scotch Plains, NJ, 2013), and other third-party process models. ESTEA2 can perform a technoeconomic analysis for a joint fermentative-catalytic process with just 12 user-supplied inputs, which, when modeled in SuperPro Designer®, required approximately eight additional inputs such as equipment design configurations. With a reduced amount of user information, ESTEA2 provides results similar to those in the literature, and more sophisticated models (ca. 7%–11% different).

**Keywords:** SuperPro; early-stage; process modeling; cost analysis; ethanol; sorbic acid

---

## 1. Introduction

Developing new biobased chemical processes is challenging, as evidenced by the multiple projects that have failed at the commercial stage [1,2]. Technoeconomic analysis at the early stages of development can reduce risks by providing a better understanding of the challenges facing a new processing system. Such analyses provide valuable technical and financial information to address project bottlenecks and better scale-up opportunities [3,4]. To perform a technoeconomic analysis using well-established, commercially available software such as SuperPro Designer® or Aspen Plus®, significant amounts of technical information related to the process are required; for example, we previously estimated that an average of six parameters per unit operation are required by SuperPro® [5]. At the early stages of process development, many of these process parameters are unknown, but this is exactly when the cost information related to the product's scope and sustainability is most needed [6]. Recognizing this paradox, and driven by the work being conducted by multiple scientists and engineers as part of the NSF Engineering Research Center for Biorenewable Chemicals (CBiRC) [7,8], Claypool and Raman developed BioPET (Biorenewables Process Evaluation Tool), an Excel-based modeling and analysis tool that can be used by researchers as they evaluate potential biobased chemical pathways [9].

BioPET was developed to perform economic analyses at the early stages of process development; as the technology matured, more complex models and simulations could be employed [9]. However, BioPET was capable of providing more detailed design and cost estimations than other preliminary models such as CAPCOST [10] or simple zero-order or proof-of-concept models that rely on extremely small parameters sets (e.g., stage yield and an estimate of the fraction of cost to feedstock). Based on feedback from academic and industrial researchers, we embarked on an effort to improve BioPET, resulting in an updated model called Early Stage Technoeconomic Analysis (ESTEA). Phase I improvements were completed by 2015 and are reported in detail, along with validation results, in the resulting MS thesis [5].

In the hope of achieving a transformation from a petrobased chemical industry to a biobased chemical industry, CBiRC is developing biobased building blocks called platform molecules capable of producing a different range of end products [8]. CBiRC's approach is a unique methodology that combines biocatalysis and chemical catalysis, thereby creating a hybrid platform technology. This method of producing biobased products can be broken down into two stages: stage 1—biocatalytic conversion of sugars to a platform molecule, stage 2—chemical catalytic conversion of the platform molecule to the desired end product [7,8,11]. The key advantage of this method is the ability to produce multiple end products from a single platform molecule; several economic benefits may arise from this approach [12]. Examples of such platform chemicals (developed by CBiRC) include triacetic acid lactone [13] and muconic acid [14]. These biologically derived products can be diversified into multiple end products through stage 2—chemical catalysis.

An example of this approach is the biocatalysis-electrocatalysis hybrid process developed by CBiRC researchers. A biological route to muconic acid from glucose, through metabolic engineering of *Saccharomyces cerevisiae* [14], was developed by one research group, while another worked on the electrocatalysis of muconic acid, producing 3-hexenedioic acid [15]. Muconic acid also serves as a platform chemical for caprolactam, adipic acid, and many other end products as well. The economic benefits of producing 3-hexenedioic acid through this technology were demonstrated using Phase I ESTEA [15].

From such case studies, additional gaps were identified in Phase I ESTEA. One of the limitations was the number of end products allowed; the tool was capable of modeling a single fermentation product, which is catalytically upgraded to a single end product. The other limitation was the number and types of downstream unit operations available. The tool provided common downstream unit operations, including decantation, adsorption, distillation, crystallization, and catalysis. However, a maximum of four downstream processing unit operations steps can be incorporated into a process. Here, we describe the improvements made to Phase I ESTEA to make Phase II ESTEA (ESTEA2) a stronger and more robust tool. The improvements undertaken focused on revising the overall structure of the model, introducing new unit operations to expand the model's capability to handle complex biobased processes, and updating the unit cost data used in the model.

The ESTEA2 model was validated using two biobased processes. We modeled a corn dry-grind-based ethanol process and then made a detailed comparison by breaking down the overall cost into its components and comparing it with several previous studies [16–18]. Also, we modeled a proposed biobased sorbic acid process. We compared these results with the results from an external vendor's sorbic acid process evaluation, again based on individual cost components.

## 2. Materials and Methods

### 2.1. ESTEA2—Model Structure

ESTEA2 is capable of modeling a single fermentation product to multiple end products, allowing a maximum of eight downstream unit operations for separation, catalytic conversion, and purification processes. Furthermore, new unit operations' (drying, batch reactor) design and cost calculations are now included in the model. The capabilities of ESTEA2 were significantly improved by these added functionalities, and streamlining the model makes it easier for users to understand the data flow.

Table 1 and Figure 1 summarize the structure and organization of ESTEA2. Based on functionality and role, the tool is divided into six brackets. The Input/Output bracket contains *Tab UI* (User Interface), which serves as the front-end platform of the entire model. Here, the user controls the primary process parameters of the model. The Design bracket will include *Tab CB* (Component Balance) and *Tab Cal* (Calculations) for computations involving process modeling. The Downstream Process Data bracket comprises *Tab EP-I* (End Product I), *Tab EP-II* (End Product II), *Tab EP-III* (End Product III), and *Tab EP-IV* (End Product IV). The *Tabs EP-I-IV* integrate process the inputs, assumptions, design, and cost calculations of downstream unit operations involved with particular end products. For example, the downstream processing information for end product-I will be grouped and represented in *Tab EP-I*. Similarly, Upstream Process Data bracket's *Tab FP* (Fermentation Process) displays all information related to the fermentation procedure. The *Tab CD* of the Cost Data bracket is responsible for computations involving cost estimations. Here the expenditure per unit kg of product produced—i.e., the Minimum Selling Price (MSP)—is computed. This MSP estimate is reported in the Final Results section in *Tab UI*. The Support bracket holds *Tab KP*, which is the tool's inventory comprising constants, unit conversions values, cost data for utilities and raw material, and process-specific assumptions. It provides unit operation specific information to the *Tab Cal* and *Tab EPs*. The other tab in the bracket is *Tab UOpS*, containing all the information related to every individual unit operation. It performs the following operations: (1) Provide process variables for selected unit operations in *Tab UI*; (2) Provide necessary process assumptions for respective unit operations from *Tab KP* to perform design calculations in *Tab Cal*; and (3) Create a data report table in *Tab EP* for every end product.

The *Tab UI* is subdivided into Plant Properties, Fermentation, and Downstream and Final Results sections. First, the user provides the plant property information, including plant operating days, internal rate of return, and plant life, and chooses a Lang Factor. Then, the user provides values for the fermentation parameters of titer, productivity, and yield along with the fermented product density. As stated earlier, ESTEA2 is capable of handling downstream processing for multiple end products by catalytically converting fermentation products. ESTEA2 can handle up to four end products, all originating from the same fermentation system. The method for designing the downstream process happens in *Tab UI*, involving the following procedure: (1) Select the number of end products; (2) Specify the annual production and density for each end product; (3) Specify downstream unit operations details (separation/catalysis). Up to eight downstream processing options can be chosen for each end product. Once the unit operation or unit type is selected, the appropriate set of process parameters is displayed, and the user enters the requisite information.

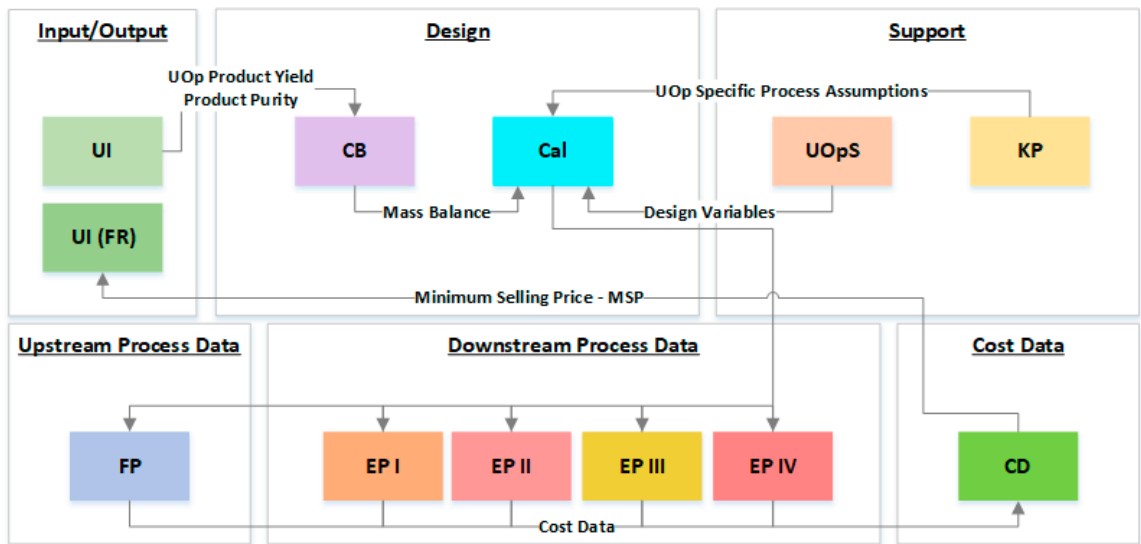

**Figure 1.** ESTEA2 Structure—Explaining Input/Output, Design, Support, Upstream, Downstream Process Data, and Cost Data brackets, their respective tabs, and flow of data across the model.

**Table 1.** Names of all tabs in ESTEA2 model and their respective key roles (acronyms used are: UI—User Interface; UI (FR)—Final Results section on User Interface; CB—Component Balance; Cal—Calculations; FP—Fermentation process; EP-I, II, III, IV—End product one, two, three, four respectively; CD—Cost Data; UOpS—Unit Operation Summary; KP—Key Parameters).

| Tab/Section Name | Roles |
| --- | --- |
| UI | User inputs including plant properties, unit operation selection, and parameters |
| UI (FR) | Displays final cost estimation result (Minimum Selling Price) |
| CB | Component balance calculations |
| Cal | Processes design calculations |
| FP | Consolidates process inputs, assumptions, design calculations; and calculates direct costs for the fermentation process |
| EP-I, II, III, IV | Consolidates process inputs, assumptions; designs calculations and calculate direct costs for the respective downstream process |
| CD | Consolidates all direct costs; computes indirect cost and minimum selling price |
| UOpS | Process information database |
| KP | Database of process assumptions, constants, unit conversions |

After all the input information is provided, ESTEA2 performs mass balance calculations in *Tab CB*. Mass and volumetric flow rates for the entire process are calculated on an hourly basis. The end product flow rate (kg/h) is calculated using operating days (user input), assuming 24-h plant operation per day. The product yield value from *Tab UI* for every procedure/process is used to back-calculate product flow out of that unit operation.

Process input parameters from *Tab UI*, mass balance data from *Tab CB*, and other required process parameters (process assumptions) from *Tab KP* are utilized to perform process model calculations, which size each unit operation. The process design and mass balance calculations are performed in *Tab Cal* and *Tab CB*, respectively, for all end products' downstream processing. Furthermore, they are methodically categorized for better understanding.

*Tab EP-I–IV* contain consolidated process details on each end product. Here, a data report is generated with information on process inputs, mass flows, process assumptions, and modeling calculation for end product downstream processing. The data report contains detailed design information subsectioned as follows: (1) Process Inputs—Unit operation-specific inputs provided by the user in *Tab UI*, (2) Process Assumptions—ESTEA2's process-specific assumptions relevant to the respective unit operation, (3) Process Flows—Mass balance data from *Tab CB*, (4) Process Calculations—Stepwise unit operation design calculation (adapted from *Tab Cal*), (5) Cost Calculations—Unit operation cost calculations (direct cost) are performed.

*2.2. Cost Calculations*

Cost calculations are performed on two different tabs. Costs directly related to unit operations (direct cost) and operating them are calculated in *Tab FP* (in the case of fermentation) and *Tab Eps* (in the case of the downstream process), whereas indirect costs are estimated in *Tab CD*. As detailed in Figure 2, the two primary components of MSP are the capital and operating costs. The capital cost is amortized to calculate the loan payment on purchased equipment costs and other construction expenses. Operating cost includes energy, labor, electricity, raw materials such as water, corn steep liquor serving as media for microbial growth, catalysts, and feedstock.

Capital cost is divided into direct and indirect costs. The direct cost is the total capital cost of all unit operations, which is the purchase equipment cost and installation cost for respective equipment. Scaling law is used to compute the raw capital cost of equipment [15]. The equation is used to calculate the equipment cost is: $C_n = (S_n/S_b)^n (C_b)$, where $C_n$—Cost of newly sized equipment, $S_n$—New size of equipment, $S_b$—Base size of equipment, $C_b$—Base cost of equipment, and n—Cost exponent. For all unit operations, ESTEA2 uses the base size, base cost, and cost component data from multiple literature

resources. The Chemical Engineering Plant Cost Index (CEPCI) values [19] are used to update the purchased equipment costs to 2018 (CEPCI for 2018: 603.1).

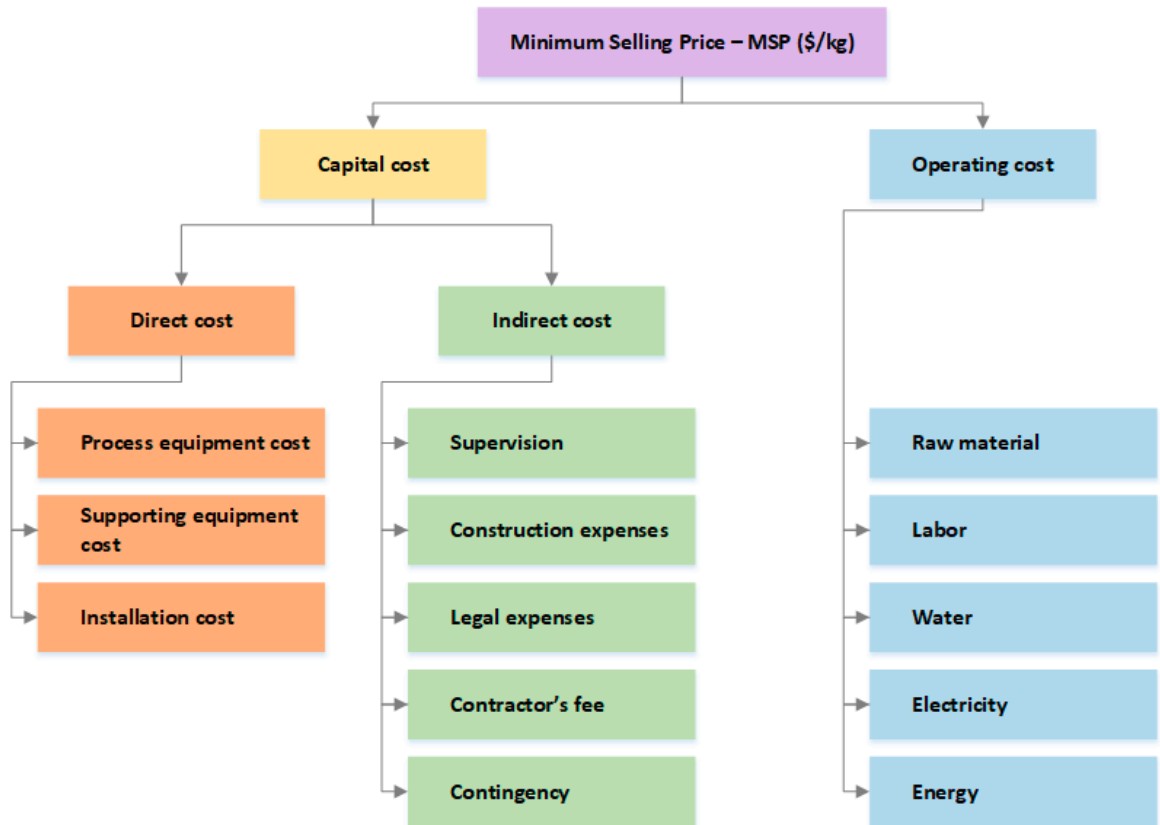

**Figure 2.** Product cost distribution elaborating direct, indirect and operating cost factors as calculated by ESTEA2.

We then use the Lang Factor method to calculate installation cost by multiplying purchased equipment cost calculated by an approximation factor. Many literature references guide Lang Factors for process plants, including [20,21]. The tool allows the user to select Lang Factors in the range of 4–10, to account for equipment installation costs as well as additional supporting equipment costs (example: solvent recovery and recycling). The direct cost is computed as the sum of purchased equipment cost and installation cost. This value is then amortized to compute annual loan payment on capital (amortized capital cost), based on the user-provided interest rate and loan period (user information).

Indirect costs account for additional expenses that are not directly related to the capital and operation cost of the plant. They are estimated as the percent of purchased equipment costs. Construction and design (34%), engineering and supervision (32%), legal expenses (4%), contractor's fee (19%), and contingency (37%) are the factors and respective percentage of purchased equipment cost used to calculate indirect cost [15].

*2.3. Unit Operation Modeling*

We evaluated process calculations related to every unit operation. Here, we discuss in detail two of the most commonly used unit operations in ESTEA2, along with two entirely new unit operations that were not in Phase-I ESTEA. The remaining five unit operations (centrifugation, decantation, extraction, crystallization, and distillation), all of which were in Phase-I ESTEA, but all of which have been enhanced, are described in the main author's [22].

### 2.3.1. Fermentation

ESTEA2 assumes a batch fermentation as the starting point of the biobased production process. Table 2 lists all process heuristics, and Table 3 lists the stepwise process design equations that are used by the fermentation process calculations. Fermentation capital and operating costs are estimated based on a small group of parameters provided by the user, along with process heuristics. The input parameters include fermented product concentration (titer in g/L), production rate (productivity in g/L/h), and yield ($kg_{Fermented\ product}/kg_{Glucose\ consumed}$). To compute the size of the fermentor, we calculate the fermentor batch time from the titer and productivity values. A standard value of 6 h of downtime is added to the batch time, to account for draining and refilling the equipment. The fermentor batch volume is computed from this batch time. A maximum equipment size of 3785 $m^3$ and base size of 757 $m^3$ (Year—2009; CEPCI—521.9) are used to calculate the total number of fermentors required. By applying the scaling law as discussed previously, the raw capital costs of newly sized fermentors are computed. Based on the calculated volumetric flow from mass balance, the number of seed fermentors required is back-calculated such that, at every seed fermentation stage, the volume increases 10-fold. The design and cost calculations of equipment and operating variables are computed in the same way as for fermentors.

Raw sugar containing glucose is considered as the feedstock, and hence the pretreatment and processing of raw materials are outside the scope of this research. Feedstock consumption is estimated as the ratio of product flow out of the fermentor to fermentation yield. These hourly data are used to calculate the annual feedstock requirements based on the total plant operating hours (POH). A feedstock unit cost of $0.14/kg is used to compute the total feedstock cost [17].

**Table 2.** The fermentation process in ESTEA2: List of user input parameters and heuristics employed by the tool.

| Parameter | Value | Reference |
|---|---|---|
| $Usable_{Fermentor}$, % | 95% | [23] |
| Max Size | 3785 $m^3$ | |
| $BS_{Fermentor}$ | 757 $m^3$ | [24] |
| $BC_{Fermentor}$ | $590,000 | [24] |
| $BS_{Seed\ Fermentor}$ | 7.57 $m^3$ | [24] |
| $BC_{Seed\ Fermentor}$ | $78,800 | [24] |
| Exp | 0.7 (dimensionless) | [24] |
| $F_{Downtime}$ | 6 h | [25] |
| $E_{R/V}$ | 15 hp/1000 gal | [26] |

**Table 3.** Fermentation process modeling in ESTEA2: Stepwise calculations as performed by the tool.

| No. | Design/Cost Equation |
|---|---|
| 1 | $TFT\ (h) = \frac{Titer\ (g/L)}{Productivity\ (g/L/h)} + F_{Downtime}\ (h)$ |
| 2 | $V_{Working,\ Batch}\ (m^3) = \frac{V_{Flow}\ (m3/h)}{Usable_{Fermentor,\ \%}}$ |
| 3 | $Number\ of\ Fermentors = \frac{V_{Working,\ Batch}\ (m^3)}{BS_{Fermentor}\ (m^3)}$ |
| 4 | $AB = \frac{POH\ (h)}{TFT\ (h)}$ |
| 5 | $Glucose\ cost\ (\$/kg) = \frac{M_{Flow,\ Product}\ (kg/h)}{F_{yield}} \times \frac{Unit\ cost_{Glucose}\ (\$/kg)}{AP\ (kg/yr) \times POH\ (h)}$ |
| 6 | $AEC\ (kWh) = E_{R/V}\ (hp/gal) \times 196.9\ (kW/m^3/hp/gal) \times V_{Working,\ Batch}\ (m^3) \times AB$ |
| 7 | $Electricity\ cost\ (\$/kg) = \frac{AEC\ (kWh) \times Unit\ cost_{Electricity}\ (\$/kWh)}{AP\ (kg/yr)}$ |

### 2.3.2. Catalysis

A multitubular reactor with the catalyst inside the tubes is considered for catalytic reactor design. The user has the privilege of selecting a catalyst out of Raney nickel, platinum, palladium, or lead

options. The user provides the selectivity of the desired product and percent conversion from the reactant to the desired product. Product solubility has to be provided to calculate the solvent flow rates. Catalyst bed porosity is computed from the bulk and particle density of catalyst chosen. The reactor volume is calculated as the product of incoming volumetric flow rate and residence time (user input). Based on the catalyst porosity, the total volume of the reactor is calculated [20]. The base size and cost of reactor used are 100 m$^2$ and $12,000 (Year—2002; CEPCI—395.6), respectively. The total catalyst required, based on a 2% catalyst loss per cycle, is computed. Like other unit operations, scaling law is used to compute purchase equipment costs. Tables S1 and S2 list the catalysis process heuristics and design calculations.

### 2.3.3. Drying

We have included spray dryer design and cost calculations in ESTEA2. Tables S3 and S4 list all the process heuristics and design calculations for spray dryer calculations performed in ESTEA2. Ambient air is heated with steam; the heated air is used to remove the water content of the product flowing in. The percentage of moisture content in the final product and product residence time provided by the user is used to design dryer size and airflow rate [27]. We assumed the air flows out at 37 °C, corresponding humidity ratio, and specific volume values from the psychrometric chart are used. The airflow rate is calculated as the ratio of the amount of water to be evaporated to the difference between humidity ratios of inlet and outlet air. Spray dryer volume is estimated from the air flowrate into the dryer and residence time. Spray dryer at 60 m$^3$ base size at $400,000 base cost (Year—2002; CEPCI—395.6) is for sizing calculations [20]. The energy consumed by the spray dryer is estimated as the amount of steam required to remove moisture/water to achieve the desired final moisture in the product. Process inputs, heuristics, and stepwise calculations are given in the Supplementary Materials.

### 2.3.4. Batch Reactor

A jacketed batch reactor design is available as one of the unit process options available in ESTEA2. This unit operation can simulate a number of batch-type processes via an assumption of residence time, including mixing, pH changes, or any other process that needs to be carried out in a batch environment. The tool offers two sizes of process vessels. The base size of 4 m$^3$ at a base cost of $30,000 (Year—2002; CEPCI—395.6) is considered as the unit size available for modeling. For a larger size, a base size of 10 m$^3$ and base cost of $50,000 are used for design and cost calculations (Table 2). This unit can be used for a procedure that depends on the residence time of mass flow, such as mixing, hydrolysis, or batch catalytic operation. Batch catalytic modeling cannot account for catalyst usage; hence, it should be calculated manually. The batch vessel is designed to accommodate a total volume for residence time specified by the user. The process heuristics and design calculations are discussed in Tables S5 and S6.

### 2.4. ESTEA2 Validation—Ethanol Process Model

We modeled an ethanol process in ESTEA2, as shown in Figure 3, and compared the results with multiple literature resources. The process was built based on Kwiatkowski's SuperPro model [18]. The ethanol process was designed for 119.1 kTA (kilotons per annum) plant capacity.

The fermentation of glucose by *Saccharomyces cerevisiae* takes place over 56 h, producing ethanol at a rate of 2 g/L/h. We assume a product yield of 51% is achieved (90% of maximum theoretical yield), producing the final concentration of ethanol at 100 g/L. A fermentor with a maximum size of 3785 m$^3$ is used, with a maximum usable percentage up to 80%. A downtime of 6 h is included to account for product discharge and cleanup.

Ethanol produced is recovered using distillation columns and molecular sieves. The first step of recovery is done by evaporating almost all of the products in distillation columns. The maximum amount of water is removed through this step. As ethanol and water form an azeotropic mixture, some water remains in ethanol after the distillation process. Molecular sieves are used as the final purification process to remove the remaining water. The smaller pores of zeolite absorb water from the ethanol-water

mixture, thereby producing 99% pure ethanol. Lang Factor of 6 is used to accommodate equipment installation as well as subsidiary equipment costs (e.g., holding tanks, heat transfer equipment).

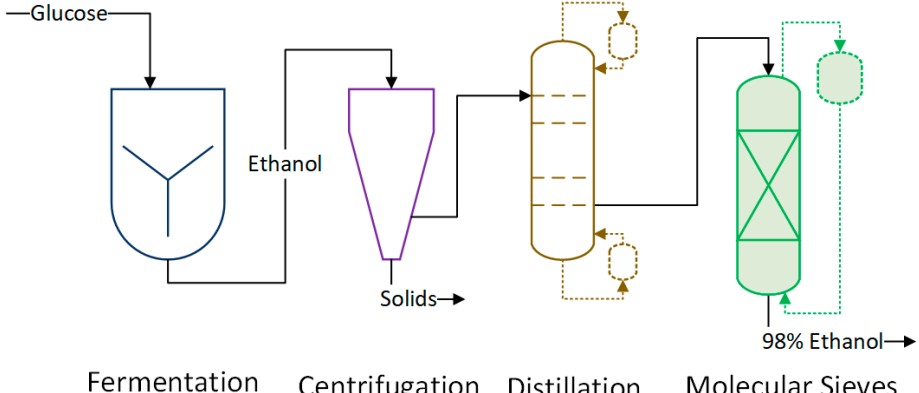

**Figure 3.** Ethanol process flow diagram as modeled in ESTEA2 (based on Kwiatkowski's SuperPro model).

No byproduct formation is assumed; hence, any design and cost calculation related to DDGS processing are not accounted for in the ethanol process modeling in ESTEA2. Similarly, pretreatment and processing of corn are not applicable to this design as glucose is used as the feedstock. Process parameters from the Hofstrand [17] and Kwiatkowski [18] models are used for modeling the fermentation and downstream processes.

*2.5. ESTEA2 Validation—Sorbic Acid Process Model*

Furthermore, the tool was validated with a postulated biobased sorbic acid process. Triacetic acid lactone (TAL), a potential platform chemical, is produced through the fermentation of sugars [28]. The biologically produced TAL is capable of undergoing chemical catalysis to form several molecules such as pogostone, dehydroacetic acid, katsumadin, acetylacetone, and sorbic acid. Sorbic acid is a high-volume commodity chemical used primarily in the food industry. Dumesic's research group has successfully produced sorbic acid from TAL through a series of chemical reactions [13]. We have utilized the available knowledge to design this hybrid process in ESTEA2 and performed its technoeconomic analysis. Since the process is still under lab-scale development, we used anticipated parametric values, instead of current values, to achieve realistic results. These data were unpublished results from CBiRC researchers working on this process (e.g., [28–30]). The results from ESTEA2 were validated by comparing with design reports from an external vendor (name not specified; referred to as EV in this work).

The sorbic acid production process was modeled in ESTEA2 for 20 kTA plant capacity. The plant is operational for 330 days per year, with 10 years of operating life. A Lang Factor of 6 is considered to account for the unit process equipment installation. Figure 4 is the process flow diagram of the sorbic acid process, which is detailed below.

Fermentation: The process begins with fermenting glucose to produce TAL. We assume a process yield of 43% (90% of theoretical yield). The fermented broth contains a TAL concentration of about 150 g/L produced at the rate of 2 g/L/h. We assume 6 h fermentor downtime for broth discharge and cleaning. We assume complete removal of cells and solid mass through centrifugation, which follows fermentation.

Toluene, Heptane, Hexanol Extraction: The purpose of this set of extraction procedures is to remove polar and nonpolar compounds, causing catalyst deactivation [29,30]. The concentration of these organic species is assumed as 1% of that of TAL. Toluene and heptane are the solvents used to remove long-chain fatty acids and other nonpolar compounds (TAL does not partition into hexane/toluene). To separate amino acids and other polar compounds, TAL in the broth is extracted

into hexanol. TAL has a partition coefficient of 7 (approximately) into 1-hexanol at low pH. Extracted TAL is passed through a silica column to remove residual polar compounds.

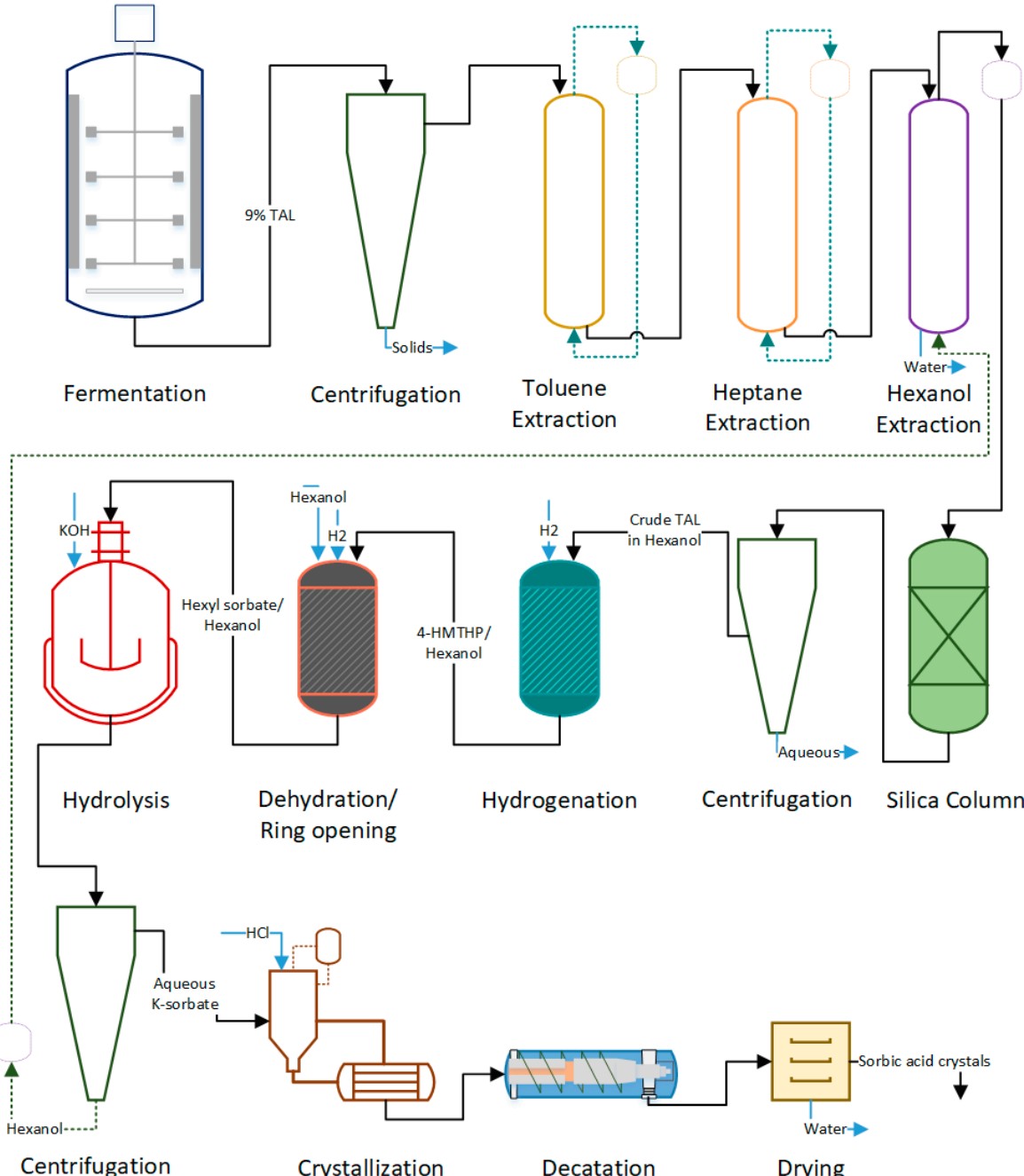

**Figure 4.** Sorbic acid process flow diagram as designed in ESTEA2 based on Chia et al., 2012 and CBiRC internal reports.

Hydrogenation: Extracted TAL in hexanol undergoes catalytic hydrogenation in the presence of Au/Pd catalyst to form 4-hydroxy-6-methyltetrahydro-2-pyrone (4-HMTHP). Steam is supplied to heat the reaction mixture to 50 °C. Natural gas, required to produce steam, is included in the energy cost calculations. We assume a 2% catalyst loss per cycle and 98% product yield during the hydrogenation process. The process, as designed in ESTEA2, uses a tubular reactor with a catalyst inside the tubes, although EV uses a stirred pressure reactor.

Dehydration/Ring Opening and Hydrolysis: 4-HMTHP undergoes dehydration at 100 °C for 12 h, followed by ring-opening at 170 °C for 12 h producing hexyl sorbate, which is hydrolyzed with KOH at a 99% conversion to K-sorbate. Dow Amberlyst—70 is the catalyst used by EV; since the catalyst is unavailable in the ESTEA2, we used Raney nickel. The overall process yield (dehydration and ring-opening together) is assumed to be 90%.

Due to limited unit operations availability, we have combined the catalysis and hydrolysis processes to occur in the same reactor. Batch reactor with a combined residence time of 26 h for dehydration, ring-opening, and hydrolysis is considered.

Crystallization and Drying: Hydrolyzed K-sorbate is later crystallized at a 98% yield. Hydrochloric acid (HCl) is used as the separating agent. The sorbic acid crystals are separated using centrifugation; the leftover aqueous phase (KCl) is recycled. Sorbic acid crystals are then dried to remove any leftover moisture. The energy required for drying sorbic acid up to 2% final moisture is computed.

## 3. Results and Discussion

Having revised Phase-I ESTEA significantly, we performed a new round of validation to ensure the tool produces accurate results. We designed biobased ethanol and sorbic acid processes in ESTEA2, as described in the earlier section. Here we detail the validation of the two models by comparing ESTEA2 results against multiple literature resources. The ethanol process was validated using multiple literature resources including ethanol profitability models (Hofstrand; Irwin) [17,31], a corn dry-grind ethanol process and cost model (Kwiatkowski) [18], ethanol production reports from USDA (Duffield; Shapouri and Gallagher) [16,32], as well as our SuperPro Designer®-based results (Claypool and Raman) [9]. The sorbic acid process was validated by comparing our results against an external vendor's (EV) detailed process design and cost estimation results for the same process. All dollar values in the data from ESTEA2 are reported in 2018 U.S. dollars.

*3.1. Ethanol Process Validation*

ESTEA2 calculated MSP for the ethanol process as $0.506/kg. Table 4 lists all the cost factors related to ethanol production. The MSP values from the literature were reported at $0.434/kg [32], $0.485/kg [31] and $0.472/kg [17] (Table 5). The overall MSP predicted by ESTEA2 was similar to the literature data, with no more than 17% variation. Feedstock cost was the dominating factor accounting for nearly 58% of the total cost. Annual operating cost (excluding feedstock) accounted for 17% and the capital cost (direct + indirect) for 25%.

**Table 4.** Cost results from Ethanol process modeling in ESTEA2—The breakdown of annual production cost and product minimum selling price.

| Cost Factor | Annual Cost | Cost per kg |
|---|---|---|
| Direct capital cost (amortized) | $4,564,125 | $0.038 |
| Indirect capital cost (amortized) | $10,791,127 | $0.090 |
| Total capital cost | $15,355,251 | $0.129 |
| Labor | $4,561,920 | $0.038 |
| Electricity | $209,969 | $0.0018 |
| Energy | $4,420,259 | $0.037 |
| Water | $460,033 | $0.004 |
| Corn steep liquor | $205,514 | $0.001 |
| Maintenance | $2,115,695 | $0.017 |
| Feedstock | $34,894,075 | $0.293 |

The literature values on amortized capital (direct) cost were $0.070/kg [31], and $0.074/kg [32] and $0.064/kg [18]. The estimations from ESTEA2 were at least 37% less than that of other literature data (Table 6). This difference is due to the exclusion of processes such as byproduct recovery, or feed processing in the case of ESTEA2.

**Table 5.** Minimum selling price comparison between ESTEA2 and literature for the dry-grind ethanol process (Acronyms used are: UI—User Interface; UI (FR)—Final Results section on User Interface; CB—Component Balance; Cal—Calculations; FP—Fermentation process; EP-I, II, III, IV—End product one, two, three, four respectively; CD—Cost Data; UOpS—Unit Operation Summary; KP—Key Parameters).

| Source | Value | ESTEA2: Source (+/−) |
|---|---|---|
| ESTEA2 | $0.506/kg | – |
| Claypool and Raman [9] | $0.521/kg | (−)3% |
| Duffield [32] | $0.434/kg | (+)17% |
| Irwin [31] | $0.485/kg | (+)4% |
| Hofstrand [17] | $0.472/kg | (+)7% |

**Table 6.** Amortized direct capital cost comparison between ESTEA2 and literature for the dry-grind ethanol process.

| Source | Value | ESTEA2: Source (+/−) |
|---|---|---|
| ESTEA2 | $0.040/kg | – |
| Claypool and Raman [9] | $0.043/kg | (−)7% |
| Duffield [32] | $0.074/kg | (−)45% |
| Irwin [31] | $0.070/kg | (−)43% |
| Kwiatkowski [18] | $0.064/kg | (−)37% |

To support this claim, Figure 5 details the distribution of $46.7 MM capital cost estimated by the Kwiatkowski model. Our estimations were 29% lower than those in Kwiatkowski's values. From the Figure, it is evident that coproduct processing, grain handling, liquefaction, and saccharification can account for 56% of total capital cost, which ESTEA2 does not include for process modeling and calculations, but some of the cost is absorbed in terms of glucose feedstock cost.

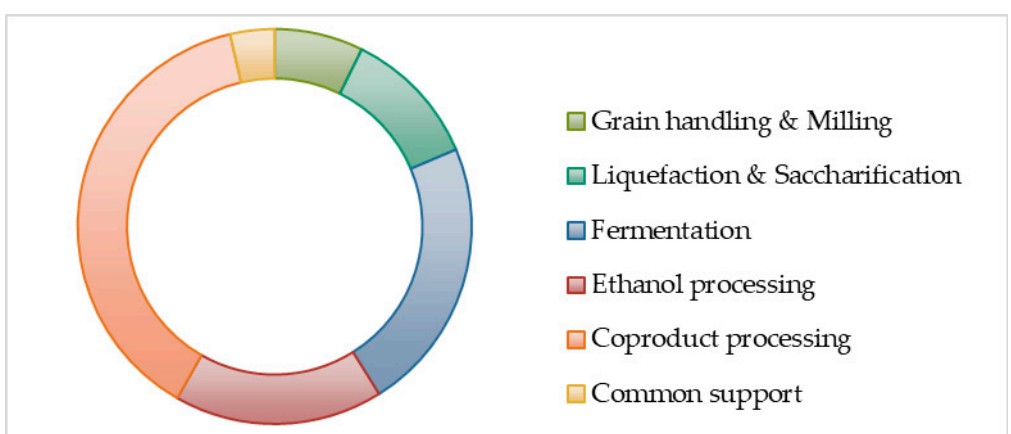

**Figure 5.** Ethanol—Capital cost distribution (based on data Kwiatkowski's SuperPro results).

Capital cost on fermentation is estimated at $0.020/kg, which was 42% more than the Kwiatkowski model ($0.014/kg ethanol) and very close to the SuperPro predicted value ($0.021/kg ethanol). Besides, ESTEA2's predictions on downstream capital costs were 43% higher. Although the fermentation and downstream processing capital costs were higher than the literature value, the overall capital cost was less. This confirms that ESTEA2's exclusion of feedstock processing, grain handling, byproduct processing, and other supporting procedures causes a difference in cost estimations. A high Lang Factor approach reduces the capital cost difference between ESTEA2 and literature and a method of handling systems with significant nonmodeled steps. This also makes ESTEA2 unsuitable for modeling

efforts that are focused on understanding pretreatment or grain handling, but that was outside of the tool's scope.

Table 7 details the operating cost comparison between ESTEA2 and other literature. Data from the Kwiatkowski model indicate that $10.3 million was spent on utilities (electricity, water, energy). The ESTEA2 model calculations were only 40% of the literature value. Results from the USDA reports [18] predict the total operating cost (including labor, maintenance, electricity, water, and energy) as $0.125/kg of ethanol produced. The ESTEA2 model's value was 8% of that of the USDA report. The operating costs variations are analyzed by studying them individually.

**Table 7.** Operating cost comparison between ESTEA2 and literature for the dry-grind ethanol process.

| Source | Value | ESTEA2: Source (+/−) |
|---|---|---|
| ESTEA2 | $0.102/kg | – |
| Claypool and Raman [9] | $0.124/kg | (−)18% |
| Duffield [32] | $0.121/kg | (−)16% |
| Kwiatkowski [18] (water, electricity, energy only) | $0.143/kg | (−)29% |

Nearly $31 million was spent annually on feedstock [18], and corn was priced at $0.087/kg. ESTEA2's feedstock, i.e., glucose cost, was $35 million, which is 13% higher than the Kwiatkowski model estimation but similar to our own SuperPro result (4% lower). The higher feedstock price in ESTEA2—$0.14/kg [18] compared to the Kwiatkowski model ($0.087/kg)—illustrates the costs involved in the pretreatment and conversion of corn to sugar.

The average thermal energy for a corn ethanol dry mill is approximately 0.03 GJ/gal [16]. On surveying 21 different ethanol plants, the energy cost per gallon of ethanol varied between $0.08 and $0.22/gallon. We considered an upper limit of $0.22/gallon for validation purposes. The ESTEA2 model's energy calculations predict that $0.044 is spent on natural gas for every kg of ethanol. This value was at least 34% lower than other literature values and 10% less than the SuperPro model result (Table S7). Again, significant energy consumption happens during the pretreatment of corn and drying of DDGS co-product processing [18]. Since ESTEA2 does not include both these operations, their cost impacts are not reflected.

The ethanol profitability model [17] assumes an electricity price of $0.081/kWh and 0.7 kWh/gallon as the ratio of electricity required per gallon of ethanol, whereas we assume an electricity price of $0.045/kWh (an approximation of the 2017 electricity price from EIA) in ESTEA2. The difference in electricity cost was the driver; as we substituted our electricity cost by $0.081/kWh, electricity costs were equivalent. We do not see a significant difference in estimations.

Our water cost predicted is 6% less than the literature value reported. We assume that 80% of water is recycled during the process, and the water cost is $0.00053/kg (Table S7). ESTEA2's water costs are 17% lower than the Kwiatkowski model and similarly to the Hofstrand model.

A labor cost comparison shows that ESTEA2's calculations are 20% higher than those of SuperPro and lower than other literature values by 26% (Table S7). This is primarily due to the dissimilar labor cost wages used by the models. The ESTEA2 model uses $25/h as the hourly wages for skilled laborers. Our model uses the following equation to calculate labor cost [10]: Labor cost = $(6.29 + 31.7P^2 + 0.23Nnp)^{0.5}$, where P = number of processing steps in handling solids and Nnp = number of processing steps in handling nonparticulate processing steps (compressors, towers, and heat exchangers).

*3.2. Sorbic Acid Process Validation*

ESTEA2 estimated the MSP of sorbic acid as $4.02/kg, while the EV's values were at $7.50/kg. As the results from the two models show a difference of $3.48/kg, we compared the pathway to MSP for both models to get a better understanding.

No capital investment was prepared as part of EV's analysis. Instead, based on plant size data (20 kTA/year), EV assumed a total capital investment of $200 MM [33]. ESTEA2 estimated the capital cost of the sorbic acid process as $106.5 MM. This is 47% lower than EV's predicted values. However, ESTEA2 assumes glucose as the feedstock; hence, the capital investments related to feedstock processing (corn to sugar) were not included as a part of this modeling effort. Furthermore, ESTEA2 eliminates any byproduct formation and, hence, respective capital investments and production costs. Additionally, unit operations such as the silica column to remove residual polar compounds and centrifuges to recover solvents were not included in the model due to unit operation availability constraints. Although not much of the information from EV's capital cost data is available, our estimation can be justified by the reasoning given above.

Throughout the process, the flow rates and annual consumption of feedstock, water, solvents, and the product do not vary more than 20% between ESTEA2 and EV. Similar flow rates imply similar equipment sizes and hence similar capital costs. For example, the fermentor size computed by ESTEA2 is only 3% larger than EV's values (fermentor volume: ESTEA2—3077 $m^3$, EV—3000 $m^3$). The total fermentation time required to achieve 90 g/L TAL concentration was calculated as 51 h (including fermentor downtime of 6 h), and that of EV was 48 h. The overall fermentation process design was very similar, without any significant outliers.

ESTEA2 underpredicts the total operating costs by 30%. The reasons, as mentioned above for capital cost differences, apply to this variation as well. To support this, we investigated the operating variables in detail as below.

ESTEA2's feedstock cost was 83% lower than EV's (Table 8). The annual feedstock consumption computed was 63 kTA (ESTEA2) and 65 kTA (EV). The glucose price used by ESTEA2 is $0.14/kg, and for EV it is $0.75/kg. The similarity between the models in terms of the amount of feedstock consumed implies that the feedstock cost used by the two models is the reason for the high difference margin, although the source of EV's feedstock price is unknown.

**Table 8.** Minimum selling price, capital, feedstock, and solvent costs comparison between ESTEA2 and EV for sorbic acid process.

| Component | Factors | EV | ESTEA2 | ESTEA2: EV |
|---|---|---|---|---|
| MSP | Cost ($/kg SA) | $7.50 | $4.02 | (−)46% |
| Capital | Cost ($/kg SA) | $200 MM | $106.5 MM | (−)47% |
| Feedstock | Unit cost ($/kg) | $0.75 | $0.14 | (−)81% |
| | Cost ($/kg SA) | $2.59 | $0.43 | (−)83% |
| | Consumption (kTA/year) | 65 | 61 | (−)6% |
| Water | Unit cost ($/kg) | $0.0009 | $0.0005 | (−)44% |
| | Cost ($/kg SA) | $0.02 | $0.004 | (−)80% |
| | Consumption (kg/year) | $3.6 \times 10^8$ | $3 \times 10^8$ | (−)17% |
| Toluene | Unit cost ($/kg) | $0.98 | $1.22 | (+)24% |
| | Cost ($/kg SA) | $0.04 | $0.06 | (+)25% |
| | Consumption (kg/year) | 3179 | 3535 | (+)11% |
| Heptane | Unit cost ($/kg) | $3.14 | $0.63 | (−)80% |
| | Cost ($/kg SA) | $0.11 | $0.03 | (−)73% |
| | Consumption (kg/year) | 3179 | 3070 | (−)4% |

Water unit costs used by ESTEA2 and EV were $0.00053/kg and $0.0009/kg, respectively (Table 8). Our tool assumes an 80% water recycling ratio, while EV's process water recycling ratio is unknown. Annual water consumption estimations were similar between the models. A difference of only 17% is observed when comparing the yearly water consumption rates. Therefore, the 80% low cost predicted by ESTEA2 is due to the difference in water unit cost.

Although heptane and toluene consumption are similar between the two models (heptane—4% variation, toluene—11% variation), their costs were significantly different. ESTEA2's toluene costs were at least 24% higher than EV's, and heptane costs were underpredicted by nearly 73%. Since the

solvent flow rates were similar, we investigated the unit cost of the solvents used by the two models. ESTEA2's heptane price was $0.63/kg, whereas that of EV was $3.14/kg. Similarly, the toluene cost was $1.22/kg in ESTEA2 and $0.985/kg in EV. These variations in unit cost data are the reflections observed in their respective operating cost differences. ESTEA2's estimated labor cost was 88% less than EV's (Table 9). Our hourly wage rate ($25/h) was $15 less than EV's. The difference in labor charges explains the differences observed in the labor cost estimation.

**Table 9.** Labor, electricity, and energy costs comparison between ESTEA2 and EV for sorbic acid.

| Component | Factors | EV | ESTEA2 | ESTEA2: EV |
|---|---|---|---|---|
| Labor | Cost ($/kg SA) | $1.11 | $0.13 | (−)88% |
| Electricity | Unit Cost ($/kWh) | $0.08 | $0.04 | (−)50% |
| | Cost ($/kg SA) | $0.068 | $0.045 | (−)34% |
| Energy (Natural gas) | Unit cost ($/MMBtu) | $4.00 | $5.54 | (+)38% |
| | Cost ($/kg SA) | $0.13 | $0.21 | (+)62% |

In the majority of process variables, the unit cost difference between ESTEA and EV have accounted for the operating cost differences. To confirm this, we replaced our unit cost data with EV's. In doing so, the dissimilarities observed in cost data were significantly reduced. Figure 6A,B illustrates the comparison between ESTEA2 and EV's estimations as a percent deviation in ESTEA2's results from EV's. Figure 6B represents the percent deviation after ESTEA2's unit cost data are replaced by the data used by EV. For instance, the MSP difference between the two results was 46%, which was improved to 22%. The MSP predicted by ESTEA2 improved from $3.34/kg to $5.85/kg (EV's MSP was $7.50/kg). The cost differences in many other process variables, including electricity, feedstock, energy, and labor costs, were reduced to less than 20%. However, EV's unit cost data do not have any literature evidence; most of their unit costs are overvalued (e.g., feedstock—glucose cost = $0.75/kg). Our data, on the other hand, are periodically updated and validated against literature resources. This analysis shows the ESTEA2's capability to model complex processes and perform detailed technoeconomic analysis by validating against a third-party external vendor's detailed report.

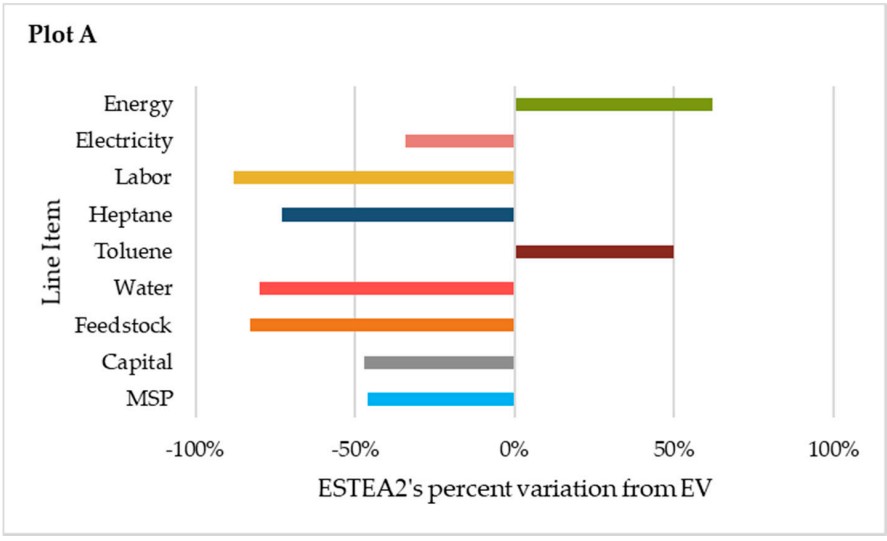

**Figure 6.** *Cont.*

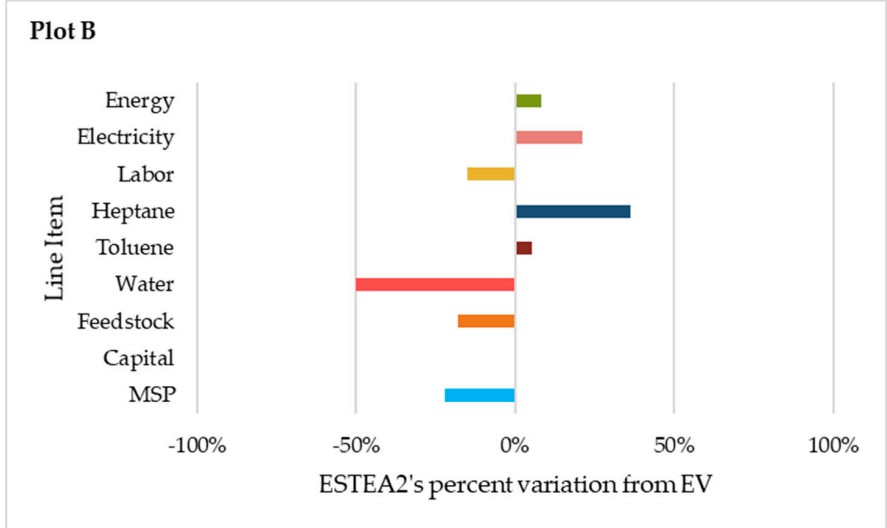

**Figure 6.** (**A**) Percent variation of ESTEA2's line-item cost results from that of EV, with our unit cost data; (**B**) Percent variation of ESTEA2's line-item cost results from that of EV, with EV's unit cost data.

## 4. Conclusions

The reorganization and advancement of ESTEA2 provide clarity in information flow and enhance the tool's ability to handle complex biobased processes. ESTEA2 validation with the ethanol process showed that the MSP predicted by the tool is 3%-17% closer to the literature results. A detailed comparison of the cost structure was performed against multiple literature resources. Furthermore, we observed only a 22% deviation in estimated MSP when compared against an external vendor's detailed technoeconomic report in the case of a sorbic acid process. These validation results show ESTEA2's strength as a process design and technoeconomic analysis platform, providing meaningful design and cost estimations.

**Supplementary Materials:** Supplementary materials can be found at http://www.mdpi.com/2227-9717/8/2/229/s1. Supplementary data of this work can be found in the online version of the paper. Table S1. Catalysis modeling in ESTEA2—User inputs and heuristics used by the tool. Table S2. Catalysis modeling in ESTEA2—stepwise calculations as performed by the tool. Table S3. Dryer modeling in ESTEA2—User inputs and heuristics used by the tool. Table S4. Dryer modeling in ESTEA2—stepwise calculations as performed by the tool. Table S5. Batch reactor modeling in ESTEA2—User inputs and heuristics used by the tool. Table S6. Batch reactor modeling in ESTEA2—stepwise calculations as performed by the tool. Table S7. Electricity, water, energy and labor cost comparison between ESTEA2 and literature for ethanol process.

**Author Contributions:** Tool construction and technoeconomic analysis, M.B.V.; Writing—Original Draft Preparation, M.B.V.; Writing—Review & Editing, D.R.R., K.A.R., and B.H.S. All authors have read and agreed to the published version of the manuscript

**Funding:** This research was funded by the National Science Foundation Engineering Center for Biorenewable Chemicals under Award No. EEC-0813570.

**Acknowledgments:** The authors are grateful to Joshua Claypool for his hard work on establishing the BioPET, which was the platform for ESTEA Phase I and ultimately for ESTEA2. The authors would like to acknowledge Professors George A. Kraus and Steven J. Hoff for their thoughtful guidance on this work. This research was funded by the National Science Foundation Engineering Center for Biorenewable Chemicals under Award No. EEC-0813570. Any opinions, findings, conclusions, and recommendations expressed in this material are those of the author(s) and do not necessarily reflect the views of the National Science Foundation.

**Conflicts of Interest:** The authors declare no conflict of interest.

## Nomenclature

| Symbol | Definition and dimension |
| --- | --- |
| AB | Annual batches in fermentation, dimensionless |
| AEC | Annual electricity consumption, kWh |
| AP | Annual production, kTA |

| | |
|---|---|
| $BC_{Fermentor}$ | Base cost of fermentor, $m^3$ |
| $BS_{Fermentor}$ | Base size of fermentor, $m^3$ |
| Cal | Calculations |
| CB | Component balance |
| CD | Cost data |
| CEPCI | Chemical Engineering Plant Cost Index |
| $E_{R/V}$ | Rate of electricity consumption in fermentation, hp/gal |
| EP | End Product |
| ESTEA2 | Early-stage technoeconomic analysis, version 2 |
| $F_{Downtime}$ | Fermentation downtime, h |
| $F_{Yield}$ | Fermentation yield, % |
| FP | Fermentation process |
| KI | Key inputs |
| KP | Key parameters |
| kTA | kilotons per annum |
| UOpS | Unit operation summary |
| $M_{Flow,Product}$ | Product mass flow, kg/h |
| MSP | Minimum selling price, ($/kg) |
| POH | Plant operating hours, h |
| TFT | Total fermentation time per batch, h |
| Unit cost$_{Glucose}$ | Cost of glucose (feedstock), $/kg |
| $V_{Flow}$ | Volumetric flow, $m^3$/h |
| $V_{Working,Batch}$ | Working volume of the batch fermentor, $m^3$ |
| Usable$_{Fermentor,\%}$ | Usable percentage of a fermentor, (%) |

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
