# Peer review of "A Technoeconomic Platform for Early-Stage Process Design and Cost Estimation of Joint Fermentative‒Catalytic Bioprocessing"

_processes, doi:10.3390/pr8020229_

Round 1
Reviewer 1 Report
This paper provides very interesting topics and is very informative. Since there are many contents to be presented in one paper, it needs logistic and clear to readers. I found currently the paper is hardly to read, the model scopes are not very clear, some details are confused.
Authors used too many “we” in describing the modelling development. Although it is acceptable, l would suggest try to avoid using first person pronouns like “we” in academic/scientific writing.
In introduction section, authors could add more introduction on joint fermentative – catalytic bioprocessing, is this model specific to glucose fermentation or any fermentation? Since fermentation is not only from glucose, will the model target to specific process or any type of bioprocessing? Without the description of the process scopes, readers may have questions like in Fermentation section Line 175: “Feedstock unit cost of $0.14/kg,” What feedstock here means? By reading later parts, I know it means glucose, but why fermentation feedstock unit cost is specific to glucose? No other feedstock can be applied to fermentation in this model?
Please correct all table formats to standard scientific writing table format.
The words in Figure 1 and Figure 2 are not clear to be seen.
Line 68-69 “The capabilities of ESTEA2 were improved significantly by this added functionality, and streamlining the model makes it easier for users to understand the data flow.” What is added functionality? Related to the first sentence of this paragraph, this sentence is not clear.
What the year of the cost based in results (Tables 4-7) ?
The describe of model structure can be improved to make it more clear to readers. Currently, it is lacking logistic, readers are hard to follow. In Figure 1, why fermentation process in process data are parallel listed with End product? Why only fermentation process in Figure 1, but in 2.3 unit operation modeling, there are also other processes like catalysis and drying not including in Figure 1? 2.3.4 Batch process description is not clear. The batch vessel is for fermentation, for distillation or for all process? Normally Batch process is in the same category of Continuous process, why authors listed it as unit operation process in the same list of fermentation? From validation examples, there are other unit process like separation (distillation, centrifugation, decantation), extraction etc., why authors only specific drying in unit operation process?
The results and discussion can be briefly, no need separate capital and operation cost in two validation cases.
Author Response
Dear Reviewer,
We warmly thank you for your time and efforts. We were able to address all your comments. In the file attached, we have included a point-by-point response (red) as well as quoted relevant edits/ new content added to the manuscript (green). Your excellent and insightful comments/questions helped us to improve the quality and clarity of our manuscript further.
Sincerely,
Mothi Viswanathan

Reviewer 2 Report
The article presented by Viswanathan et al. describes a tool for the techno-economic evaluation of industrial bioprocesses involving fermentation or fermentation plus catalytic conversion of biomass. The main difference between this tool and established software for the same purpose is the reduced need for data input by the user, allowing the TE evaluation to take place earlier in the process design cycle. This is a sound study which can benefit from some minor improvements:
abstract: SuperPro is referred to as a sophisticated tool but it's really not in comparison to Aspen, for example. Please rephrase first sentence. section 2.1: I understand from this section and the results for the ethanol process that ESTEA2 does not consider the pre-treatment steps. Yet in the results section you were able to show basic calculations with regards to the energy input required. How could these calculation be integrated in ESTEA2? In addition, there is no cost/dedicated units for microorganism growth prior to the production fermenter. Bacteria/yeast cannot go straight from frozen into that reactor scale. Figures 1 and 2 are very blurry Section 2.2: what's the plan for maintaining the capital cost calculations current? is there built-in inflation calculator? Figure 4. typo: crystallization The entries in the last column of Tables 5 and 6 should read lower or higher not low or high. Better depicted as +/- rather than in words. Please also amend associated text. The work on parametric cost would be better described as a sensitivity analysis.Author Response
Dear Reviewer,
We warmly thank you for your time and efforts. We were able to address all your comments. In the file attached, we have included a point-by-point response (red) as well as quoted relevant edits/ new content added to the manuscript (green). Your excellent and insightful comments/questions helped us to improve the quality and clarity of our manuscript further.
Sincerely,
Mothi Viswanathan

Round 2
Reviewer 1 Report
Thanks the authors for thorough editing on the paper to improve its logic and clarity.
Authors need clarify which year of cost this study based on? 2017, 2018, or 2019 US value or other years? I tried to find this kind of information but couldn't find from the paper. Authors need add one sentence something like "All dollar values are reported in 20xx U.S. dollars."
I see authors answered "Due to the challenges of accounting for technological changes, we did not attempt to correct values for consumer price index." But still software Superpro Designer give you an input of which year of your analysis is started? You need specific this, otherwise, how other readers refer to your results as any comparison? Because a dollar in 2010 is not equal to a dollar in 2020. If you used different year values, you need convert them into same year, then make further analysis.
Thanks.
Author Response
Dear Reviewer,
Please find the attached response word file.
Best,
Mothi

Round 3
Reviewer 1 Report
Thanks for authors to make all changes recording to the reviewer's comments.